# MULTI-AGENT INTERACTIONS MODELING WITH CORRELATED POLICIES

**Minghuan Liu**[1]**, Ming Zhou**[1]**, Weinan Zhang** [1]**, Yuzheng Zhuang**[2]**, Jun Wang**[2]**, Wulong Liu**[2]**, Yong Yu**[1]
[1] Shanghai Jiaotong University, [2] Huawei Noah's Ark Lab
{minghuanliu, mingak, wnzhang, yyu}@sjtu.edu.cn, {zhuangyuzheng, w.j, liuwulong}@huawei.com

## ABSTRACT

In multi-agent systems, complex interacting behaviors arise due to the high correlations among agents. However, previous work on modeling multi-agent interactions from demonstrations is primarily constrained by assuming the independence among policies and their reward structures. In this paper, we cast the multi-agent interactions modeling problem into a multi-agent imitation learning framework with explicit modeling of correlated policies by approximating opponents' policies, which can recover agents' policies that can regenerate similar interactions. Consequently, we develop a Decentralized Adversarial Imitation Learning algorithm with Correlated policies (CoDAIL), which allows for decentralized training and execution. Various experiments demonstrate that CoDAIL can better regenerate complex interactions close to the demonstrators and outperforms state-of-the-art multi-agent imitation learning methods. Our code is available at `https://github.com/apexrl/CoDAIL`.

## 1 INTRODUCTION

Modeling complex interactions among intelligent agents from the real world is essential for understanding and creating intelligent multi-agent behaviors, which is typically formulated as a multi-agent learning (MAL) problem in multi-agent systems. When the system dynamics are agnostic and non-stationary due to the adaptive agents with implicit goals, multi-agent reinforcement learning (MARL) is the most commonly used technique for MAL. MARL has recently drawn much attention and achieved impressive progress on various non-trivial tasks, such as multi-player strategy games (OpenAI, 2018; Jaderberg et al., 2018), traffic light control (Chu et al., 2019), taxi-order dispatching (Li et al., 2019) etc.

A central challenge in MARL is to specify a good learning goal, as the agents' rewards are correlated and thus cannot be maximized independently (Bu et al., 2008). Without explicit access to the reward signals, imitation learning could be the most intuitive solution for learning good policies directly from demonstrations. Conventional solutions such as behavior cloning (BC) (Pomerleau, 1991) learn the policy in a supervised manner by requiring numerous data while suffering from compounding error (Ross & Bagnell, 2010; Ross et al., 2011). Inverse reinforcement learning (IRL) (Ng et al., 2000; Russell, 1998) alleviates these shortcomings by recovering a reward function but is always expensive to obtain the optimal policy due to the forward reinforcement learning procedure in an inner loop. Generative adversarial imitation learning (GAIL) (Ho & Ermon, 2016) leaves a better candidate for its model-free structure without compounding error, which is highly effective and scalable. However, real-world multi-agent interactions could be much challenging to imitate because of the strong correlations among adaptive agents' policies and rewards. Consider if a football coach wants to win the league, he must make targeted tactics against various opponents, in addition to the situation of his team. Moreover, the multi-agent environment tends to give rise to more severe compounding errors with more expensive running costs.

Motivated by these challenges, we investigate the problem of modeling complicated multi-agent interactions from a pile of off-line demonstrations and recover their on-line policies, which can regenerate analogous multi-agent behaviors. Prior studies for multi-agent imitation learning typically limit the complexity in demonstrated interactions by assuming isolated reward structures (Barrett et al., 2017; Le et al., 2017; Lin et al., 2014; Waugh et al., 2013) and independence in per-agent

policies that overlook the high correlations among agents (Song et al., 2018; Yu et al., 2019). In this paper, we cast the multi-agent interactions modeling problem into a multi-agent imitation learning framework with correlated policies by approximating opponents' policies, in order to reach inaccessible opponents' actions due to concurrently execution of actions among agents when making decisions. Consequently, with approximated opponents model, we develop a Decentralized Adversarial Imitation Learning algorithm with Correlated policies (CoDAIL) suitable for learning correlated policies under our proposed framework, which allows for decentralized training and execution. We prove that our framework treats the demonstrator interactions as one of $\epsilon$-Nash Equilibrium ($\epsilon$-NE) solutions under the recovered reward.

In experiments, we conduct multi-dimensional comparisons for both the reward gap between learned agents and demonstrators, along with the distribution divergence between demonstrations and regenerated interacted trajectories from learned policies. Furthermore, the results reveal that CoDAIL can better recover correlated multi-agent policy interactions than other state-of-the-art multi-agent imitation learning methods in several multi-agent scenarios. We further illustrate the distributions of regenerated interactions, which indicates that CoDAIL yields the closest interaction behaviors to the demonstrators.

## 2 PRELIMINARIES

### 2.1 MARKOV GAME AND $\epsilon$-NASH EQUILIBRIUM

Markov game (MG), or stochastic game (Littman, 1994), can be regarded as an extension of Markov Decision Process (MDP). Formally, we define an MG with $N$ agents as a tuple $\langle N, \mathcal{S}, \mathcal{A}^{(1)}, \dots, \mathcal{A}^{(N)}, P, r^{(1)}, \dots, r^{(N)}, \rho_0, \gamma \rangle$, where $\mathcal{S}$ is the set of states, $\mathcal{A}^{(i)}$ represents the action space of agent $i$, where $i \in \{1, 2, \dots, N\}$, $P : \mathcal{S} \times \mathcal{A}^{(1)} \times \mathcal{A}^{(2)} \times \cdots \times \mathcal{A}^{(N)} \times \mathcal{S} \to \mathbb{R}$ is the state transition probability distribution, $\rho_0 : \mathcal{S} \to \mathbb{R}$ is the distribution of the initial state $s^0$, and $\gamma \in [0, 1]$ is the discounted factor. Each agent $i$ holds its policy $\pi^{(i)}(a^{(i)}|s) : \mathcal{S} \times \mathcal{A}^{(i)} \to [0, 1]$ to make decisions and receive rewards defined as $r^{(i)} : \mathcal{S} \times \mathcal{A}^{(1)} \times \mathcal{A}^{(2)} \times \cdots \times \mathcal{A}^{(N)} \to \mathbb{R}$. We use $-i$ to represent the set of agents except $i$, and variables without superscript $i$ to denote the concatenation of all variables for all agents (e.g., $\pi$ represents the joint policy and $a$ denotes actions of all agents). For an arbitrary function $f : \langle s, a \rangle \to \mathbb{R}$, there is a fact that $\mathbb{E}_\pi[f(s, a)] = \mathbb{E}_{s \sim P, a \sim \pi}[f(s, a)] \triangleq \mathbb{E}\left[\sum_{t=0}^\infty \gamma^t f(s_t, a_t)\right]$, where $s^0 \sim \rho_0$, $a_t \sim \pi$, $s_{t+1} \sim P(s_{t+1}|a_t, s_t)$. The objective of agent $i$ is to maximize its own total expected return $R^{(i)} \triangleq \mathbb{E}_\pi[r^{(i)}(s, a)] = \mathbb{E}\left[\sum_{t=0}^\infty \gamma^t r^{(i)}(s_t, a_t)\right]$.

In Markov games, however, the reward function for each agent depends on the joint agent actions. Such a fact implies that one's optimal policy must also depend on others' policies. For the solution to the Markov games, $\epsilon$-Nash equilibrium ($\epsilon$-NE) is a commonly used concept that extends Nash equilibrium (NE) (Nash, 1951).

**Definition 1.** *An $\epsilon$-NE is a strategy profile $(\pi_*^{(i)}, \pi_*^{(-i)})$ such that $\exists \epsilon > 0$:*

$$v^{(i)}(s, \pi_*^{(i)}, \pi_*^{(-i)}) \geq v^{(i)}(s, \pi^{(i)}, \pi_*^{(-i)}) - \epsilon, \forall \pi^{(i)} \in \Pi^{(i)} , \tag{1}$$

*where $v^{(i)}(s, \pi^{(i)}, \pi^{(-i)}) = \mathbb{E}_{\pi^{(i)}, \pi^{(-i)}, s_0 = s}\left[r^{(i)}(s_t, a_t^{(i)}, a_t^{(-i)})\right]$ is the value function of agent $i$ under state $s$, and $\Pi^{(i)}$ is the set of policies available to agent $i$.*

$\epsilon$-NE is weaker than NE, which can be seen as sub-optimal NE. Every NE is equivalent to an $\epsilon$-NE where $\epsilon = 0$.

### 2.2 GENERATIVE ADVERSARIAL IMITATION LEARNING

Imitation learning aims to learn the policy directly from expert demonstrations without any access to the reward signals. In single-agent settings, such demonstrations come from behavior trajectories sampled with the expert policy, denoted as $\tau_E = \{(s_t, a_t^{(i)})\}_{t=0}^\infty$. However, in multi-agent settings, demonstrations are often interrelated trajectories, that is, which are sampled from the interactions of policies among all agents, denoted as $\Omega_E = \{(s_t, a_t^{(1)}, \dots, a_t^{(N)})\}_{t=0}^\infty$. For simplicity, we will use the term *interactions* directly as the concept of interrelated trajectories, and we refer to trajectories for a single agent.

Typically, behavior cloning (BC) and inverse reinforcement learning (IRL) are two main approaches for imitation learning. Although IRL theoretically alleviates compounding error and outperforms to BC, it is less efficient since it requires resolving an RL problem inside the learning loop. Recent proposed work aims to learn the policy without estimating the reward function directly, notably, GAIL (Ho & Ermon, 2016), which takes advantage of Generative Adversarial Networks (GAN (Goodfellow et al., 2014)), showing that IRL is the dual problem of occupancy measure matching. GAIL regards the environment as a black-box, which is non-differentiable but can be leveraged through Monte-Carlo estimation of policy gradients. Formally, its objective can be expressed as

$$\min_{\pi} \max_{D} \mathbb{E}_{\pi_E} \left[ \log D(s,a) \right] + \mathbb{E}_{\pi} \left[ \log \left( 1 - D(s,a) \right) \right] - \lambda H(\pi) , \tag{2}$$

where $D$ is a discriminator that identifies the expert trajectories with agents' sampled from policy $\pi$, which tries to maximize its evaluation from $D$; $H$ is the causal entropy for the policy and $\lambda$ is the hyperparameter.

## 2.3 Correlated Policy

In multi-agent learning tasks, each agent $i$ makes decisions independently while the resulting reward $r^{(i)}(s_t, a_t^{(i)}, a_t^{(-i)})$ depends on others' actions, which makes its cumulative return subjected to the joint policy $\pi$. One common joint policy modeling method is to decouple the $\pi$ with assuming conditional independence of actions from different agents (Albrecht & Stone, 2018):

$$\pi(a^{(i)}, a^{(-i)}|s) \approx \pi^{(i)}(a^{(i)}|s)\pi^{(-i)}(a^{(-i)}|s) . \tag{3}$$

However, such a non-correlated factorization on the joint policy is a vulnerable simplification which ignores the influence of opponents (Wen et al., 2019). And the learning process of agent $i$ lacks stability since the environment dynamics depends on not only the current state but also the joint actions of all agents (Tian et al., 2019). To solve this, recent work has taken opponents into consideration by decoupling the joint policy as a correlated policy conditioned on state $s$ and $a^{(-i)}$ as

$$\pi(a^{(i)}, a^{(-i)}|s) = \pi^{(i)}(a^{(i)}|s, a^{(-i)})\pi^{(-i)}(a^{(-i)}|s) , \tag{4}$$

where $\pi^{(i)}(a^{(i)}|s, a^{(-i)})$ is the conditional policy, with which agent $i$ regards all potential actions from its opponent policies $\pi^{(-i)}(a^{(-i)}|s)$, and makes decisions through the marginal policy $\pi^{(i)}(a^{(i)}|s) = \int_{a^{(-i)}} \pi^{(i)}(a^{(i)}|s, a^{(-i)})\pi^{(-i)}(a^{(-i)}|s)\, \mathrm{d}a^{(-i)} = \mathbb{E}_{a^{(-i)}} \pi^{(i)}(a^{(i)}|s, a^{(-i)})$.

## 3 Methodology

### 3.1 Generalize Correlated Policies to Multi-Agent Imitation Learning

In multi-agent settings, for agent $i$ with policy $\pi^{(i)}$, it seeks to maximize its cumulative reward against demonstrator opponents who equip with demonstrated policies $\pi_E^{(-i)}$ via reinforcement learning:

$$\mathrm{RL}^{(i)}(r^{(i)}) = \arg\max_{\pi^{(i)}} \lambda H(\pi^{(i)}) + \mathbb{E}_{\pi^{(i)}, \pi_E^{(-i)}}[r^{(i)}(s, a^{(i)}, a^{(-i)})] , \tag{5}$$

where $H(\pi^{(i)})$ is the $\gamma$-discounted entropy (Bloem & Bambos, 2014; Haarnoja et al., 2017) of policy $\pi^{(i)}$ and $\lambda$ is the hyperparameter. By coupling with Eq. (5), we define an IRL procedure to find a reward function $r^{(i)}$ such that the demonstrated joint policy outperforms all other policies, with the regularizer $\psi : \mathbb{R}^{\mathcal{S} \times \mathcal{A}^{(1)} \times \cdots \times \mathcal{A}^{(N)}} \to \overline{\mathbb{R}}$:

$$\begin{aligned}
\mathrm{IRL}_\psi^{(i)}(\pi_E^{(i)}) = \arg\max_{r^{(i)}} -\psi(r^{(i)}) - \max_{\pi^{(i)}}(\lambda H(\pi^{(i)}) + \mathbb{E}_{\pi^{(i)}, \pi_E^{(-i)}}[r^{(i)}(s, a^{(i)}, a^{(-i)})]) \\
+ \mathbb{E}_{\pi_E}[r^{(i)}(s, a^{(i)}, a^{(-i)})] .
\end{aligned} \tag{6}$$

It is worth noting that we cannot obtain the demonstrated policies from the demonstrations directly. To solve this problem, we first introduce the occupancy measure, namely, the unnormalized distribution of $\langle s, a \rangle$ pairs correspond to the agent interactions navigated by joint policy $\pi$:

$$\rho_\pi(s, a) = \pi(a|s) \sum_{t=0}^{\infty} \gamma^t P(s_t = s|\pi) . \tag{7}$$

With the definition in Eq. (7), we can further formulate $\rho_\pi$ from agent $i$'s perspective as

$$
\begin{aligned}
\rho_\pi(s, a^{(i)}, a^{(-i)}) &= \pi(a^{(i)}, a^{(-i)}|s) \sum_{t=0}^{\infty} \gamma^t P(s_t = s|\pi^{(i)}, \pi^{(-i)}) \\
&= \rho_{\pi^{(i)}, \pi^{(-i)}}(s, a^{(i)}, a^{(-i)}) \\
&= \begin{cases} \underbrace{\pi^{(i)}(a^{(i)}|s)\pi^{(-i)}(a^{(-i)}|s)}_{\text{non-correlated form}} \sum_{t=0}^{\infty} \gamma^t P(s_t = s|\pi^{(i)}, \pi^{(-i)}) \\ \underbrace{\pi^{(i)}(a^{(i)}|s, a^{(-i)})\pi^{(-i)}(a^{(-i)}|s)}_{\text{correlated form}} \sum_{t=0}^{\infty} \gamma^t P(s_t = s|\pi^{(i)}, \pi^{(-i)}) \end{cases},
\end{aligned}
\tag{8}
$$

where $a^{(i)} \sim \pi^{(i)}$ and $a^{(-i)} \sim \pi^{(-i)}$. Furthermore, with the support of Eq. (8), we have

$$
\begin{aligned}
\mathbb{E}_{\pi^{(i)}, \pi^{(-i)}}[\cdot] &= \mathbb{E}_{s \sim P, a^{(i)} \sim \pi^{(i)}}[\mathbb{E}_{a^{(-i)} \sim \pi^{(-i)}}[\cdot]] \\
&= \sum_{s, a^{(i)}, a^{(-i)}} \rho_{\pi^{(i)}, \pi^{(-i)}}(s, a^{(i)}, a^{(-i)})[\,\cdot\,].
\end{aligned}
\tag{9}
$$

In analogy to the definition of occupancy measure of that in a single-agent environment, we follow the derivation from Ho & Ermon (2016) and state the conclusion directly[1].

**Proposition 1.** *The IRL regarding demonstrator opponents is a dual form of a occupancy measure matching problem with regularizer $\psi$, and the induced optimal policy is the primal optimum, specifically, the policy learned by RL on the reward recovered by IRL can be characterize by the following equation:*

$$\text{RL}^{(i)} \circ \text{IRL}^{(i)} = \underset{\pi^{(i)}}{\arg\min} -\lambda H(\pi^{(i)}) + \psi^*(\rho_{\pi^{(i)}, \pi_E^{(-i)}} - \rho_{\pi_E}) . \tag{10}$$

With setting the regularizer $\psi = \psi_{GA}$ similar to Ho & Ermon (2016), we can obtain a GAIL-like imitation algorithm to learn $\pi_E^{(i)}$ from $\pi_E$ given demonstrator counterparts $\pi_E^{(-i)}$ by introducing the adversarial training procedures of GANs which lead to a saddle point $(\pi^{(i)}, D^{(i)})$:

$$\min_{\pi^{(i)}} \max_{D^{(i)}} -\lambda H(\pi^{(i)}) + \mathbb{E}_{\pi_E}\left[\log D^{(i)}(s, a^{(i)}, a^{(-i)})\right] + \mathbb{E}_{\pi^{(i)}, \pi_E^{(-i)}}\left[\log(1 - D^{(i)}(s, a^{(i)}, a^{(-i)}))\right],$$

$$\tag{11}$$

where $D^{(i)}$ denotes the discriminator for agent $i$, which plays a role of surrogate cost function and guides the policy learning.

However, such an algorithm is not practical, since we are unable to access the policies of demonstrator opponents $\pi_E^{(-i)}$ because the demonstrated policies are always given through sets of interactions data. To alleviate this deficiency, it is necessary to deal with accessible counterparts. Thereby we propose Proposition 2.

**Proposition 2.** *Let $\mu$ be an arbitrary function such that $\mu$ holds a similar form as $\pi^{(-i)}$, then*

$$\mathbb{E}_{\pi^{(i)}, \pi^{(-i)}}[\cdot] = \mathbb{E}_{\pi^{(i)}, \mu}\left[\frac{\rho_{\pi^{(i)}, \pi^{(-i)}}(s, a^{(i)}, a^{(-i)})}{\rho_{\pi^{(i)}, \mu}(s, a^{(i)}, a^{(-i)})} \cdot \right].$$

*Proof.* Substituting $\pi^{(-i)}$ with $\mu$ in Eq. (9) by importance sampling. □

---

[1]Note that Ho & Ermon (2016) proved the conclusion under the goal to minimize the cost instead of maximizing the reward of an agent.

Proposition 2 raises an important point that a term of importance weight can quantify the demonstrator opponents. By replacing $\pi_E^{(-i)}$ with $\pi^{(-i)}$, Eq. (11) is equivalent with

$$\min_{\pi^{(i)}} \max_{D^{(i)}} -\lambda H(\pi^{(i)}) + \mathbb{E}_{\pi_E} \left[ \log D^{(i)}(s, a^{(i)}, a^{(-i)}) \right] + \mathbb{E}_{\pi^{(i)}, \pi^{(-i)}} \left[ \alpha \log \left(1 - D^{(i)}(s, a^{(i)}, a^{(-i)})\right) \right],$$
(12)

where $\alpha = \frac{\rho_{\pi^{(i)}, \pi_E^{(-i)}}(s, a^{(i)}, a^{(-i)})}{\rho_{\pi^{(i)}, \pi^{(-i)}}(s, a^{(i)}, a^{(-i)})}$ is the importance sampling weight. In practice, it is challenging to estimate the densities and the learning methods might suffer from large variance. Thus, we fix $\alpha = 1$ in our implementation, and as the experimental results have shown, it has no significant influences on performance. Besides, a similar approach can be found in Kostrikov et al. (2018).

So far, we've built a multi-agent imitation learning framework, which can be easily generalized to correlated or non-correlated policy settings. No prior has to be considered in advance since the discriminator is able to learn the implicit goal for each agent.

## 3.2 LEARN WITH THE OPPONENTS MODEL

With the objective shown in Eq. (11), demonstrated interactions can be imitated by updating discriminators to offer surrogate rewards and learning their policies alternately. Formally, the update of discriminator for each agent $i$ can be expressed as:

$$\nabla_\omega J_D(\omega) = \mathbb{E}_{s \sim P, a^{(-i)} \sim \pi^{(-i)}} \left[ \int_{a^{(i)}} \pi_\theta^{(i)}(a^{(i)}|s, a^{(-i)}) \nabla_\omega \log \left(1 - D_\omega^{(i)}(s, a^{(i)}, a^{(-i)})\right) \mathrm{d}a^{(i)} \right]$$
$$+ \mathbb{E}_{(s, a^{(i)}, a^{(-i)}) \sim \Omega_E} \left[ \nabla_\omega \log D_\omega^{(i)}(s, a^{(i)}, a^{(-i)}) \right],$$
(13)

and the update of policy is:

$$\nabla_\theta J_\pi(\theta) = \mathbb{E}_{s \sim P, a^{(-i)} \sim \pi^{(-i)}} \left[ \nabla_{\theta^{(i)}} \int_{a^{(i)}} \pi_\theta^{(i)}(a^{(i)}|s, a^{(-i)}) A^{(i)}(s, a^{(i)}, a^{(-i)}) \mathrm{d}a^{(i)} \right] - \lambda \nabla_{\theta^{(i)}} H(\pi_\theta^{(i)}),$$
(14)

where discriminator $D^{(i)}$ is parametrized by $\omega$, and the policy $\pi^{(i)}$ is parametrized by $\theta$. It is worth noting that the agent $i$ considers opponents' action $a^{(-i)}$ while updating its policy and discriminator, with integrating all its possible decisions to find the optimal response. However, it is unrealistic to have the access to opponent joint policy $\pi(a^{(-i)}|s)$ for agent $i$. Thus, it is possible to estimate opponents' actions via approximating $\pi^{(-i)}(a^{(-i)}|s)$ using opponent modeling. To that end, we construct a function $\sigma^{(i)}(a^{(-i)}|s) : \mathcal{S} \times \mathcal{A}^{(1)} \times \cdots \times \mathcal{A}^{(i-1)} \times \mathcal{A}^{(i+1)} \times \cdots \times \mathcal{A}^{(N)} \to [0,1]^{N-1}$, as the approximation of opponents for each agent $i$. Then we rewrite Eq. (13) and Eq. (14) as:

$$\nabla_\omega J_D(\omega) \approx \mathbb{E}_{s \sim P, \hat{a}^{(-i)} \sim \sigma^{(i)}, a^{(i)} \sim \pi_\theta^{(i)}} \left[ \nabla_{\omega^{(i)}} \log(1 - D_\omega^{(i)}(s, a^{(i)}, \hat{a}^{(-i)})) \right]$$
$$+ \mathbb{E}_{(s, a^{(i)}, a^{(-i)}) \sim \Omega_E} \left[ \nabla_\omega \log D_\omega^{(i)}(s, a^{(i)}, a^{(-i)}) \right]$$
(15)

and

$$\nabla_\theta J_\pi(\theta) \approx \mathbb{E}_{s \sim P, \hat{a}^{(-i)} \sim \sigma^{(i)}, a^{(i)} \sim \pi_\theta^{(i)}} \left[ \nabla_{\theta^{(i)}} \log \pi_\theta^{(i)}(a^{(i)}|s, \hat{a}^{(-i)}) A^{(i)}(s, a^{(i)}, \hat{a}^{(-i)}) \right] - \lambda \nabla_{\theta^{(i)}} H(\pi_\theta^{(i)})$$
(16)

respectively. Therefore, each agent $i$ must infer the opponents model $\sigma^{(i)}$ to approximate the unobservable policies $\pi^{(-i)}$, which can be achieved via supervised learning. Specifically, we learn in discrete action space by minimizing a cross-entropy (CE) loss, and a mean-square-error (MSE) loss in continuous action space:

$$L = \begin{cases} \frac{1}{2} \mathbb{E}_{s \sim p} \left[ \left\| \sigma^{(i)}(a^{(-i)}|s) - \pi^{(-i)}(a^{(-i)}|s) \right\|^2 \right], & \text{continuous action space} \\ \mathbb{E}_{s \sim p} \left[ \pi^{(-i)}(a^{(-i)}|s) \log \sigma^{(i)}(a^{(-i)}|s) \right], & \text{discrete action space.} \end{cases}$$
(17)

With opponents modeling, agents are able to be trained in a fully decentralized manner. We name our algorithm as Decentralized Adversarial Imitation Learning with Correlated policies (Correlated DAIL, a.k.a. CoDAIL) and present the training procedure in Appendix Algo. 1, which can be easily scaled to a distributed algorithm. As a comparison, we also present a non-correlated DAIL algorithm with non-correlated policy assumption in Appendix Algo. 2.

### 3.3 THEORETICAL ANALYSIS

In this section, we prove that the reinforcement learning objective against demonstrator counterparts shown in the last section is essentially equivalent to reaching an $\epsilon$-NE.

Since we fix the policies of agents $-i$ as $\pi_E^{(-i)}$, the RL procedure mentioned in Eq. (5) can be regarded as a single-agent RL problem. Similarly, with a fixed $\pi_E^{(-i)}$, the IRL process of Eq. (6) is cast to a single-agent IRL problem, which recovers an optimal reward function $r_*^{(i)}$ which achieves the best performance following the joint action $\pi_E$. Thus we have

$$\begin{aligned}
\text{RL}^{(i)}(r_*^{(i)}) &= \underset{\pi^{(i)}}{\arg\max} \; \lambda H(\pi^{(i)}) + \mathbb{E}_{\pi^{(i)}, \pi_E^{(-i)}}[r^{(i)}(s, a^{(i)}, a^{(-i)})] \\
&= \pi_E^{(i)} \; .
\end{aligned} \tag{18}$$

We can also rewrite Eq. (18) as

$$\lambda H(\pi_E^{(i)}) + \mathbb{E}_{\pi_E^{(i)}, \pi_E^{(-i)}}[r^{(i)}(s, a^{(i)}, a^{(-i)})] \geq \lambda H(\pi^{(i)}) + \mathbb{E}_{\pi^{(i)}, \pi_E^{(-i)}}[r^{(i)}(s, a^{(i)}, a^{(-i)})] \tag{19}$$

for all $\pi^{(i)} \in \Pi^{(i)}$, which is equivalent to

$$\begin{aligned}
\mathbb{E}_{a_t^{(i)} \sim \pi_E^{(i)}, a_t^{(-i)} \sim \pi_E^{(-i)}, s_0 = s} &\left[ \sum_{t=0}^{\infty} \gamma^t r_*^{(i)}(s_t, a_t^{(i)}, a_t^{(-i)}) \right] \geq \\
\mathbb{E}_{a_t^{(i)} \sim \pi^{(i)}, a_t^{(-i)} \sim \pi_E^{(-i)}, s_0 = s} &\left[ \sum_{t=0}^{\infty} \gamma^t r_*^{(i)}(s_t, a_t^{(i)}, a_t^{(-i)}) \right] + \lambda(H(\pi^{(i)}) - H(\pi_E^{(i)})), \forall \pi^{(i)} \in \Pi^{(i)} \; .
\end{aligned} \tag{20}$$

Given the value function defined in Eq. (1) for each agent $i$, for $H(\pi^{(i)}) - H(\pi_E^{(i)}) < 0, \forall \pi^{(i)} \in \Pi^{(i)}$, we have

$$v^{(i)}(s, \pi_E^{(i)}, \pi_E^{(-i)}) \geq v^{(i)}(s, \pi^{(i)}, \pi_E^{(-i)}) - \lambda(H(\pi_E^{(i)}) - H(\pi^{(i)})) \; . \tag{21}$$

For $H(\pi^{(i)}) - H(\pi_E^{(i)}) \geq 0, \forall \pi^{(i)} \in \Pi^{(i)}$ we have

$$\begin{aligned}
v^{(i)}(s, \pi_E^{(i)}, \pi_E^{(-i)}) &\geq v^{(i)}(s, \pi^{(i)}, \pi_E^{(-i)}) + \lambda(H(\pi^{(i)}) - H(\pi_E^{(i)})) \\
&\geq v^{(i)}(s, \pi^{(i)}, \pi_E^{(-i)}) - \lambda(H(\pi^{(i)}) - H(\pi_E^{(i)})) \; .
\end{aligned} \tag{22}$$

Let $\epsilon = \lambda \max \left\{ \left| H(\pi^{(i)}) - H(\pi_E^{(i)}) \right|, \forall \pi^{(i)} \in \Pi^{(i)} \right\}$, then we finally obtain

$$v^{(i)}(s, \pi_E^{(i)}, \pi_E^{(-i)}) \geq v^{(i)}(s, \pi^{(i)}, \pi_E^{(-i)}) - \epsilon, \forall \pi^{(i)} \in \Pi^{(i)} \; , \tag{23}$$

which is exactly the $\epsilon$-NE defined in Definition 1. We can always prove that $\epsilon$ is bounded in small values such that the $\epsilon$-NE solution concept is meaningful. Generally, random policies that keep vast entropy are not always considered as sub-optimal solutions or demonstrated policies $\pi_E^{(i)}$ in most reinforcement learning environments. As we do not require those random policies, we can remove them from the candidate policy set $\Pi^{(i)}$, which indicates that $H(\pi^{(i)})$ is bounded in small values, so as $\epsilon$. Empirically, we adopt a small $\lambda$, and attain the demonstrator policy $\pi_E$ with an efficient learning algorithm to become a close-to-optimal solution.

Thus, we conclude that the objective of our CoDAIL assumes that demonstrated policies institute an $\epsilon$-NE solution concept (but not necessarily unique) that can be controlled the hyperparameter $\lambda$ under some specific reward function, from which the agent learns a policy. It is worth noting that Yu et al. (2019) claimed that NE is incompatible with maximum entropy inverse reinforcement learning (MaxEnt IRL) because NE assumes that the agent never takes sub-optimal actions. Nevertheless, we prove that given demonstrator opponents, the multi-agent MaxEnt IRL defined in Eq. (6) is equivalent to finding an $\epsilon$-NE.

## 4    RELATED WORK

Albeit non-correlated policy learning guided by a centralized critic has shown excellent properties in couple of methods, including MADDPG (Lowe et al., 2017), COMA (Foerster et al., 2018), MA Soft-Q (Wei et al., 2018), it lacks in modeling complex interactions because its decisions making relies on the independent policy assumption which only considers private observations while ignores the impact of opponent behaviors. To behave more rational, agents must take other agents into consideration, which leads to the studies of opponent modeling (Albrecht & Stone, 2018) where an agent models how its opponents behave based on the interaction history when making decisions (Claus & Boutilier, 1998; Greenwald et al., 2003; Wen et al., 2019; Tian et al., 2019).

For multi-agent imitation learning, however, prior works fail to learn from complicated demonstrations, and many of them are bounded with particular reward assumptions. For instance, Bhattacharyya et al. (2018) proposed Parameter Sharing Generative Adversarial Imitation Learning (PS-GAIL) that adopts parameter sharing trick to extend GAIL to handle multi-agent problems directly, but it does not utilize the properties of Markov games with strong constraints on the action space and the reward function. Besides, there are many works built-in Markov games that are restricted under tabular representation and known dynamics but with specific prior of reward structures, as fully cooperative games (Barrett et al., 2017; Le et al., 2017; Šošic et al., 2016; Bogert & Doshi, 2014), two-player zero-sum games (Lin et al., 2014), two-player general-sum games (Lin et al., 2018), and linear combinations of specific features (Reddy et al., 2012; Waugh et al., 2013).

Recently, some researchers take advantage of GAIL to solve Markov games. Inspired by a specific choice of Lagrange multipliers for a constraint optimization problem (Yu et al., 2019), Song et al. (2018) derived a performance gap for multi-agent from NE. It proposed multi-agent GAIL (MA-GAIL), where they formulated the reward function for each agent using private actions and observations. As an improvement, Yu et al. (2019) presented a multi-agent adversarial inverse reinforcement learning (MA-AIRL) based on logistic stochastic best response equilibrium and MaxEnt IRL. However, both of them are inadequate to model agent interactions with correlated policies with independent discriminators. By contrast, our approach can generalize correlated policies to model the interactions from demonstrations and employ a fully decentralized training procedure without to get access to know the specific opponent policies.

Except for the way of modeling multi-agent interactions as recovering agents' policies from demonstrations, which can regenerate similar interacted data, some other works consider different effects of interactions. Grover et al. (2018) proposed to learn a policy representation function of the agents based on their interactions and sets of generalization tasks using the learned policy embeddings. They regarded interactions as the episodes that contain only $k$ (in the paper they used 2 agents), which constructs an agent-interaction graph. Different from us, they focused on the potential relationships among agents to help characterize agent behaviors. Besides, Kuhnt et al. (2016) and Gindele et al. (2015) proposed to use the Dynamic Bayesian Model that describes physical relationships among vehicles and driving behaviors to model interaction-dependent behaviors in autonomous driving scenario.

Correlated policy structures that can help agents consider the influence of other agents usually need opponents modeling (Albrecht & Stone, 2018) to infer others' actions. Opponent modeling has a rich history in MAL (Billings et al., 1998; Ganzfried & Sandholm, 2011), and lots of researches have recently worked out various useful approaches for different settings in deep MARL, e.g., DRON (He et al., 2016) and ROMMEO (Tian et al., 2019). In this paper, we focus on imitation learning with correlated policies, and we choose a natural and straightforward idea of opponent modeling that learning opponents' policies in the way of supervised learning with historical trajectories. Opponent models are used both in the training and the execution stages.

## 5    EXPERIMENTS

### 5.1    EXPERIMENTAL SETTINGS

**Environment Description**    We test our method on the Particle World Environments (Lowe et al., 2017), which is a popular benchmark for evaluating multi-agent algorithms, including several cooperative and competitive tasks. Specifically, we consider two cooperative scenarios and two com-

Table 1: Average reward gaps between demonstrators and learned agents in 2 cooperative tasks. Means and standard deviations are taken across different random seeds.

| Algorithm | Coop.-Comm. | Coop.-Navi. |
|---|---|---|
| Demonstrators | $0 \pm 0$ | $0 \pm 0$ |
| MA-AIRL | $0.780 \pm 0.917$ | $6.696 \pm 3.646$ |
| MA-GAIL | $0.638 \pm 0.624$ | $7.596 \pm 3.088$ |
| NC-DAIL | $0.692 \pm 0.597$ | $6.912 \pm 3.971$ |
| CoDAIL | $\mathbf{0.632 \pm 0.685}$ | $\mathbf{6.249 \pm 2.779}$ |
| Random | $186.001 \pm 16.710$ | $322.1124 \pm 15.358$ |

Table 2: Average reward gaps between demonstrators and learned agents in 2 competitive tasks, where 'agent+' and 'agent-' represent 2 teams of agents and 'total' is their sum. Means and standard deviations are taken across different random seeds.

| Algorithm | Keep-away | | | Pred.-Prey | | |
|---|---|---|---|---|---|---|
| | Total | Agent+ | Agent- | Total | Agent+ | Agent- |
| Demonstrators | $0 \pm 0$ | $0 \pm 0$ | $0 \pm 0$ | $0 \pm 0$ | $0 \pm 0$ | $0 \pm 0$ |
| MA-AIRL | $12.273 \pm 1.817$ | $4.149 \pm 1.912$ | $8.998 \pm 4.345$ | $279.535 \pm 77.903$ | $35.100 \pm 1.891$ | $174.235 \pm 73.168$ |
| MA-GAIL | $1.963 \pm 1.689$ | $1.104 \pm 1.212$ | $1.303 \pm 0.798$ | $15.788 \pm 10.887$ | $4.800 \pm 2.718$ | $8.826 \pm 3.810$ |
| NC-DAIL | $1.805 \pm 1.695$ | $1.193 \pm 0.883$ | $1.539 \pm 1.188$ | $27.611 \pm 14.645$ | $8.260 \pm 7.087$ | $6.975 \pm 5.130$ |
| CoDAIL | $\mathbf{0.269 \pm 0.078}$ | $\mathbf{0.064 \pm 0.041}$ | $\mathbf{0.219 \pm 0.084}$ | $\mathbf{10.456 \pm 6.762}$ | $\mathbf{4.500 \pm 3.273}$ | $\mathbf{4.359 \pm 2.734}$ |
| Random | $28.272 \pm 2.968$ | $25.183 \pm 2.150$ | $53.455 \pm 2.409$ | $100.736 \pm 6.870$ | $37.980 \pm 2.396$ | $13.204 \pm 8.444$ |

petitive ones as follows: 1) Cooperative-communication, with 2 agents and 3 landmarks, where an unmovable speaker knowing the goal, cooperates with a listener to reach a particular landmarks who achieves the goal only through the message from the speaker; 2) Cooperative-navigation, with 3 agents and 3 landmarks, where agents must cooperate via physical actions and it requires each agent to reach one landmark while avoiding collisions; 3) Keep-away, with 1 agent, 1 adversary and 1 landmark, where the agent has to get close to the landmark, while the adversary is rewarded by pushing away the agent from the landmark without knowing the target; 4) Predator-prey, with 1 prey agent with 3 adversary predators, where the slower predator agents must cooperate to chase the prey agent that moves faster and try to run away from the adversaries.

**Experimental Details**    We aim to compare the quality of interactions modeling in different aspects. To obtain the interacted demonstrations sampled from correlated policies, we train the demonstrator agent via a MARL learning algorithm with opponents modeling to regard others' policies into one's decision making, since the ground-truth reward in those simulated environments is accessible. Specifically, we modify the multi-agent version ACKTR (Wu et al., 2017; Song et al., 2018), an efficient model-free policy gradient algorithm, by keeping an auxiliary opponents model and a conditioned policy for each agent, which can transform the original centralized on-policy learning algorithm to be decentralized. Note that we do not necessarily need experts that can do well in our designated environments. Instead, any demonstrator will be treated as it is from an $\epsilon$-NE strategy concept under some unknown reward functions, which will be recovered by the discriminator. In our training procedure, we first obtain demonstrator policies induced by the ground-truth rewards and then generate demonstrations, i.e., the interactions data for imitation training. Then we train the agents through the surrogate rewards from discriminators. We compare CoDAIL with MA-AIRL, MA-GAIL, non-correlated DAIL (NC-DAIL) (the only difference between MA-GAIL and NC-DAIL is whether the reward function depends on joint actions or individual action) and a random agent. We do not apply any prior to the reward structure for all tasks to let the discriminator learn the implicit goals. All training procedures are pre-trained via behavior cloning to reduce the sample complexity, and we use 200 episodes of demonstrations, each with a maximum of 50 timesteps.

## 5.2 REWARD GAP

Tab. 1 and Tab. 2 show the averaged absolute differences of reward for learned agents compared to the demonstrators in cooperative and competitive tasks, respectively. The learned interactions are considered superior if there are smaller reward gaps. Since cooperative tasks are reward-sharing, we show only a group reward for each task in Tab. 1. Compared to the baselines, CoDAIL achieves smaller gaps in both cooperative and competitive tasks, which suggests that our algorithm has a

Table 3: KL divergence of learned agents position distribution and demonstrators position distribution from an individual perspective in different scenarios. 'Total' is the KL divergence for state-action pairs of all agents, and 'Per' is the averaged KL divergence of each agent. Experiments are conducted under the same random seed. Note that unmovable agents are not recorded since they never move from the start point, and there is only one movable agent in Cooperative-communication.

| Algorithm | Coop.-Comm. | Coop.-Navi. | | Keep-away | | Pred.-Prey | |
|---|---|---|---|---|---|---|---|
| | Total/Per | Total | Per | Total | Per | Total | Per |
| Demonstrators | 0 | 0 | 0 | 0 | 0 | 0 | 0 |
| MA-AIRL | 35.525 | 18.071 | 47.241 | 69.146 | 98.248 | 71.568 | 118.511 |
| MA-GAIL | 34.681 | 15.034 | 45.550 | 51.721 | 69.820 | 9.998 | 27.116 |
| NC-DAIL | 38.002 | 16.202 | 46.040 | 46.563 | 61.780 | 16.698 | 33.307 |
| CoDAIL | **6.427** | **9.033** | **23.100** | **3.113** | **5.735** | **8.621** | **22.600** |
| Random | 217.456 | 174.892 | 221.209 | 191.344 | 234.829 | 37.555 | 82.361 |

robust imitation learning capability of modeling the demonstrated interactions. It is also worth noting that CoDAIL achieves higher performance gaps in competitive tasks than cooperative ones, for which we think that conflict goals motivate more complicated interactions than a shared goal. Besides, MA-GAIL and NC-DAIL are about the same, indicating that less important is the surrogate reward structure on these multi-agent scenarios. To our surprise, MA-AIRL does not perform well in some environments, and even fails in Predator-prey. We list the raw obtained rewards in Appendix C, and we provide more hyperparameter sensitivity results in Appendix D.

## 5.3 DIVERGENCE OVER INTERACTIONS

Since we aim to recover the interactions of agents generated by the learned policies, it is proper to evaluate the relevance between distributions of regenerated interactions and demonstration data. Specifically, we collect positions of agents over hundreds of state-action tuples, which can be regarded as the low-dimension projection of the state-action interactions. We start each episode from a different initial state but the same for each algorithm in one episode. We run all the experiments under the same random seed, and collect positions of each agent in the total 100 episodes, each with a maximum of 50 timesteps.

We first estimate the distribution of position $(x, y)$ via Kernel Density Estimation (KDE) (Rosenblatt, 1956) with Gaussian kernel to compute the Kullback-Leibler (KL) divergence between the generated interactions with the demonstrated ones, shown in Tab. 3. It is evident that in terms of the KL divergence between regenerated interactions with demonstrator interactions, CoDAIL generates the interaction data that obtains the minimum gap with the demonstration interaction, and highly outperforms other baseline methods. Besides, MA-GAIL and NC-DAIL reflect about-the-same performance to model complex interactions, while MA-AIRL behaves the worst, even worse than random agents on Predator-prey.

## 5.4 VISUALIZATIONS OF INTERACTIONS

To further understand the interactions generated by learned policies compared with the demonstrators, we visualize the interactions for demonstrator policies and all learned ones. We plot the density distribution of positions, $(x, y)$ and marginal distributions of $x$-position and $y$-position. We illustrate the results conducted on Keep-away in Fig. 1, other scenarios can be found in the Appendix E. Higher frequency positions in collected data are colored darker in the plane, and higher the value with respect to its marginal distributions.

As shown in Fig. 1, the interaction densities of demonstrators and CoDAIL agents are highly similar (and with the smallest KL divergence), which tend to walk in the right-down side. In contrast, other learned agents fail to recover the demonstrator interactions. It is worth noting that even different policies can interact to earn similar rewards, but still keep vast differences among their generated interactions. Furthermore, such a result reminds us that the real reward is not the best metric to evaluate the quality of modeling the demonstrated interactions or imitation learning (Li et al., 2017).

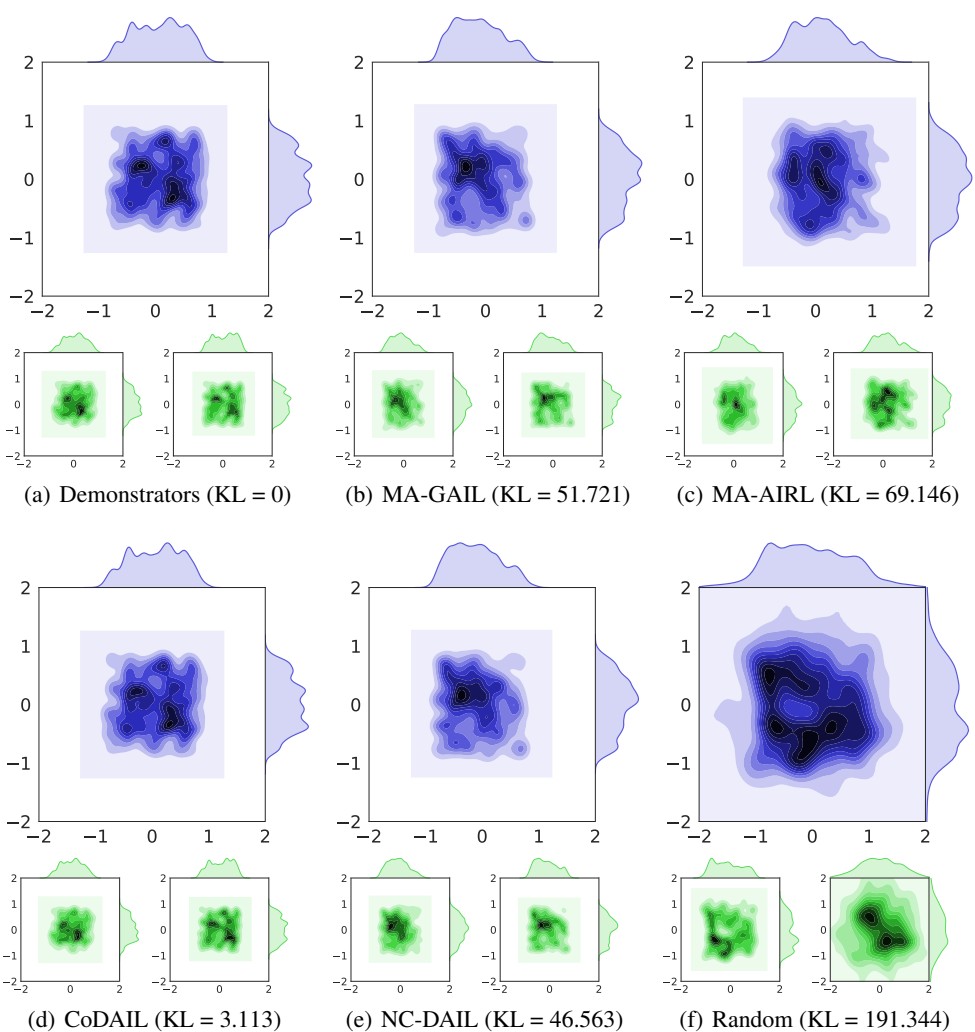

Figure 1: The density and marginal distribution of agent positions, in 100 repeated episodes with different initialized states, generated from different learned policies upon Keep-away. The top row of each sub-figure is drawn from state-action pairs of all agents. Meanwhile, the bottom row explains for each individual (KL means the KL divergence between generated interactions shown in the top row and the demonstrators).

## 6 CONCLUSION

In this paper, we focus on modeling complex multi-agent interactions via imitation learning on demonstration data. We develop a decentralized adversarial imitation learning algorithm with correlated policies (CoDAIL) with approximated opponents modeling. CoDAIL allows for decentralized training and execution and is more capable of modeling correlated interactions from demonstrations shown by multi-dimensional comparisons against other state-of-the-art multi-agent imitation learning methods on several experiment scenarios. In the future, we will consider covering more imitation learning tasks and modeling the latent variables of policies for diverse multi-agent imitation learning.

## ACKNOWLEDGEMENT

We sincerely thank Yaodong Yang for helpful discussion. The corresponding author Weinan Zhang is supported by NSFC (61702327, 61772333, 61632017). The author Minghuan Liu is supported by Wu Wen Jun Honorary Doctoral Scholarship, AI Institute, Shanghai Jiao Tong University.

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

# Appendices

## A ALGORITHM OUTLINES

### A.1 CODAIL ALGORITHM

Algo. 1 demonstrates the outline for our CoDAIL algorithm with non-correlated policy structure defined in Eq. (4), where we approximate the opponents model $\sigma^{(i)}(a^{(-i)}|s)$, improve the discriminator $D^{(i)}$ and the policy $\pi^{(i)}$ iteratively.

---

**Algorithm 1** CoDAIL Algorithm

---

1: **Input:** Expert interactive demonstrations $\Omega_E \sim \pi_E$, $N$ policy parameters $\theta^{(1)}, ..., \theta^{(N)}$, $N$ value parameters $\phi^{(1)}, ..., \phi^{(N)}$, $N$ opponents models parameters $\psi^{(1)}, ..., \psi^{(N)}$ and $N$ discriminator parameters $\omega^{(1)}, ..., \omega^{(N)}$;

2: **for** $k = 0, 1, 2, \ldots$ **do**

3:     Sample interactions among $N$ agents $\Omega_k \sim \pi$, where $\pi$ is in correlated policy structure and each agent $i$ utilizes its opponents model $\sigma^{(i)}$ to help making decisions.

4:     **for** agent $i = 1, 2, \ldots, N$ **do**

5:         Use state-action pairs $(s, a^{(-i)}) \in \Omega_k$ to update $\psi^{(i)}$ to minimize the objective as shown in Eq. (17).

6:         For every state-action pair $(s, a^{(i)}) \in \Omega_k$, sample estimated opponent policies from opponents model: $\hat{a}^{(-i)} \sim \sigma^{(i)}(a^{(i)}|s)$, and update $\omega^{(i)}$ with the gradient as shown in Eq. (15).

7:         Compute advantage estimation $A^{(i)}$ for each tuple $(s, a^{(i)}, , \hat{a}^{(-i)})$ with surrogate reward function $r^{(i)}(s, a^{(i)}, \hat{a}^{(-i)}) = \log(D^{(i)}_{\omega^{(i)}}(s, a^{(i)}, \hat{a}^{(-i)})) - \log(1 - D^{(i)}_{\omega^{(i)}}(s, a^{(i)}, \hat{a}^{(-i)}))$

$$A^{(i)}(s_t, a_t^{(i)}, \hat{a}_t^{(-i)}) = \sum_{k=0}^{T-1}(\gamma^t r^{(i)}(s_{t+k}, a_{t+k}^{(i)}, \hat{a}_{t+k}^{(-i)})) + \gamma^T V^{(i)}_{\phi^{(i)}}(s, a_{T-1}^{(-i)}) \quad (24)$$

$$- V^{(i)}_{\phi^{(i)}}(s, a_{t-1}^{(-i)}) \quad (25)$$

8:         Update $\phi^{(i)}$ to minimize the objective:

$$L(\phi^{(i)}) = \left\| \sum_{t=0}^{T} \gamma^t r^{(i)}(s, a_t^{(i)}, a_t^{(-i)}) - \hat{V}^{(i)}(s_t, a_{t-1}^{(-i)}) \right\|^2 \quad (26)$$

9:         Update $\theta^{(i)}$ following the gradient shown in Eq. (16):

$$\mathbb{E}_{s\sim P, \hat{a}^{(-i)}\sim\sigma^{(i)}, a^{(i)}\sim\pi_\theta^{(i)}} \left[ \nabla_{\theta^{(i)}} \log \pi_\theta^{(i)}(a^{(i)}|s, \hat{a}^{(-i)}) A^{(i)}(s, a^{(i)}, \hat{a}^{(-i)}) \right] - \lambda \nabla_{\theta^{(i)}} H(\pi_\theta^{(i)}) \quad (27)$$

10:     **end for**

11: **end for**

---

### A.1.1 NC-DAIL ALGORITHM

We outline the step by step NC-DAIL algorithm with non-correlated decomposition of joint policy defined in Eq. (3) in Algo. 2.

---

**Algorithm 2** NC-DAIL Algorithm

---

1: **Input:** Expert interactive demonstrations $\Omega_E \sim \pi_E$, $N$ policy parameters $\theta^{(1)}, ..., \theta^{(N)}$, $N$ value parameters $\phi^{(1)}, ..., \phi^{(N)}$ and $N$ discriminator parameters $\omega^{(1)}, ..., \omega^{(N)}$;

2: **for** $k = 0, 1, 2, \ldots$ **do**

3:      Sample interactions between $N$ agents $\Omega_k \sim \pi$, where $\pi$ is in non-correlated policy structure.

4:      **for** agent $i = 1, 2, \ldots, N$ **do**

5:          Use $(s, a^{(i)}, a^{(-i)}) \in \Omega_k$ to update $\omega^{(i)}$ with the gradient:

$$\hat{\mathbb{E}}_{\Omega_k}\left[\nabla_\omega \log(D^{(i)}_{\omega^{(i)}}(s, a^{(i)}, a^{(-i)}))\right] + \hat{\mathbb{E}}_{\Omega_E}[\nabla_{\omega^{(i)}} \log(D^{(i)}_{\omega^{(i)}}(s, a^{(i)}, a^{(-i)}))] . \quad (28)$$

6:          Compute advantage estimation $A^{(i)}$ for $(s, a^{(i)}, a^{(-i)}) \in \Omega_k$ with surrogate reward function $r^{(i)}(s, a^{(i)}, a^{(-i)}) = \log(D^{(i)}_{\omega^{(i)}}(s, a^{(i)}, a^{(-i)})) - \log(1 - D^{(i)}_{\omega^{(i)}}(s, a^{(i)}, a^{(-i)}))$

$$A^{(i)}(s_t, a_t^{(i)}, a_t^{(-i)}) = \sum_{k=0}^{T-1}(\gamma^t r^{(i)}(s_{t+k}, a_{t+k}^{(i)}, a_{t+k}^{(-i)})) + \gamma^T V^{(i)}_{\phi^{(i)}}(s, a_{T-1}^{(-i)}) \quad (29)$$

$$- V^{(i)}_{\phi^{(i)}}(s, a_{t-1}^{(-i)}) \quad (30)$$

7:          Update $\phi^{(i)}$ to minimize the objective:

$$L(\phi^{(i)}) = \left\|\sum_{t=0}^{T} \gamma^t r^{(i)}(s, a_t^{(i)}, a_t^{(-i)}) - \hat{V}^{(i)}(s_t, a_{t-1}^{(-i)})\right\|^2 \quad (31)$$

8:          Update $\theta^{(i)}$ by taking a gradient step with:

$$\hat{\mathbb{E}}_{\Omega_k}\left[\nabla_{\theta^{(i)}} \log \pi^{(i)}(a^{(i)}, s) A^{(i)}(s, a)\right] - \lambda \nabla_{\theta^{(i)}} H(\pi_\theta^{(i)}) . \quad (32)$$

9:      **end for**

10: **end for**

---

## B MODEL ARCHITECTURES

During our experiments, we use two layer MLPs with 128 cells in each layer, for policy networks, value networks, discriminator networks and opponents model networks on all scenarios. Specifically for opponents models, we utilize a multi-head-structure network, where each head predicts each opponent's action separately, and we get the overall opponents joint action $a^{(-i)}$ by concatenating all actions. The batch size is set to 1000. The policy is trained using K-FAC optimizer (Martens & Grosse, 2015) with learning rate of 0.1 and with a small $\lambda$ of 0.05. All other parameters for K-FAC optimizer are the same in (Wu et al., 2017). We train each algorithm for 55000 epochs with 5 random seeds to gain its average performance on all environments.

## C  RAW RESULTS

We list the raw obtained rewards of all algorithms in each scenarios.

Table 4: Raw average total rewards in 2 comparative tasks. Means and standard deviations are taken across different random seeds.

| Algorithm | Coop.-Comm. | Coop.-Navi. |
|---|---|---|
| Demonstrators | -24.560 ± 1.213 | -178.597 ± 6.383 |
| MA-AIRL | -25.366 ± 1.492 | -172.733 ± 5.595 |
| MA-GAIL | -25.081 ± 1.421 | -172.169 ± 4.105 |
| NC-DAIL | -25.177 ± 1.371 | -171.685 ± 4.591 |
| CoDAIL | -25.107 ± 1.486 | -183.846 ± 5.728 |
| Random | -247.606 ± 17.842 | -1139.569 ± 19.192 |

Table 5: Raw average rewards of each agent in 2 competitive tasks, where agent+ and agent- represent 2 teams of agents and total is their sum. Means and standard deviations are taken across different random seeds.

| Algorithm | Keep-away | | |
|---|---|---|---|
| | Total | Agent+ | Agent- |
| Demonstrators | -18.815 ± 0.909 | -12.092 ± 0.617 | -6.723 ± 0.430 |
| MA-AIRL | -31.088 ± 2.371 | -15.367 ± 3.732 | -15.721 ± 4.448 |
| MA-GAIL | -20.778 ± 0.994 | -12.818 ± 1.105 | -7.959 ± 0.796 |
| NC-DAIL | -20.619 ± 0.957 | -12.357 ± 1.424 | -8.262 ± 1.310 |
| CoDAIL | -19.084 ± 0.882 | -12.142 ± 0.578 | -6.942 ± 0.433 |
| Random | -47.086 ± 2.485 | 13.091 ± 2.032 | -60.177 ± 2.225 |
| Algorithm | Pred.-Prey | | |
| | Total | Agent+ | Agent- |
| Demonstrators | 65.202 ± 18.661 | 44.820 ± 4.663 | -69.258 ± 5.361 |
| MA-AIRL | -210.546 ± 80.333 | 8.040 ± 3.626 | -234.666 ± 71.165 |
| MA-GAIL | 65.202 ± 18.661 | 44.820 ± 4.663 | -69.258 ± 5.361 |
| NC-DAIL | 59.553 ± 30.684 | 42.320 ± 10.323 | -67.407 ± 3.700 |
| CoDAIL | 79.445 ± 5.913 | 47.480 ± 4.067 | -61.909 ± 6.367 |
| Random | -31.747 ± 7.865 | 5.160 ± 1.170 | -47.227 ± 7.830 |

# D HYPERPARAMETER SENSITIVITY

Table 6: Results of different training frequency (1:4, 1:2, 1:1, 2:1, 4:1) of $D$ and $G$ on Communication-navigation. Means and standard deviations are taken across different random seeds.

| Training Frequency | Total Reward Difference |
|---|---|
| 1:4 | $2541.144 \pm 487.711$ |
| 1:2 | $12.004 \pm 5.496$ |
| 1:1 | $6.249 \pm 2.779$ |
| 2:1 | $1136.255 \pm 1502.604$ |
| 4:1 | $2948.878 \pm 1114.528$ |

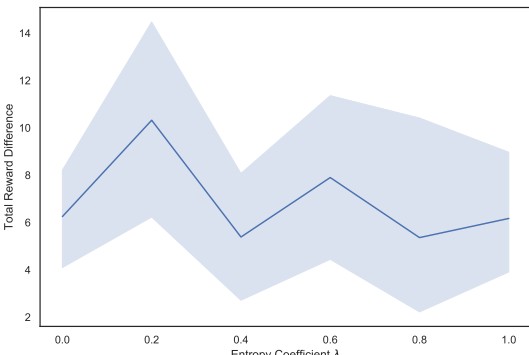

Figure 2: Results of different entropy coefficient $\lambda$.

We evaluate how the stability of our algorithm when the hyperparameters change during our experiments on Communication-navigation. Tab. 6 shows the total reward difference between learned agents and demonstrators when we modify the training frequency of $D$ and $G$ (i.e., the policy), which indicates that the frequencies of $D$ and $G$ are more stable when $D$ is trained slower than $G$, and the result reaches a relative better performance when the frequency is 1:2 or 1:1. Fig. 2 illustrates that the choice of $\lambda$ has little effect on the total performance. The reason may be derived from the discrete action space in this environment, where the policy entropy changes gently.

# E INTERACTION VISUALIZATIONS UPON OTHER SCENARIOS

We show the density of interactions for different methods along with demonstrator policies conducted upon Cooperative-communication in Fig. 3.

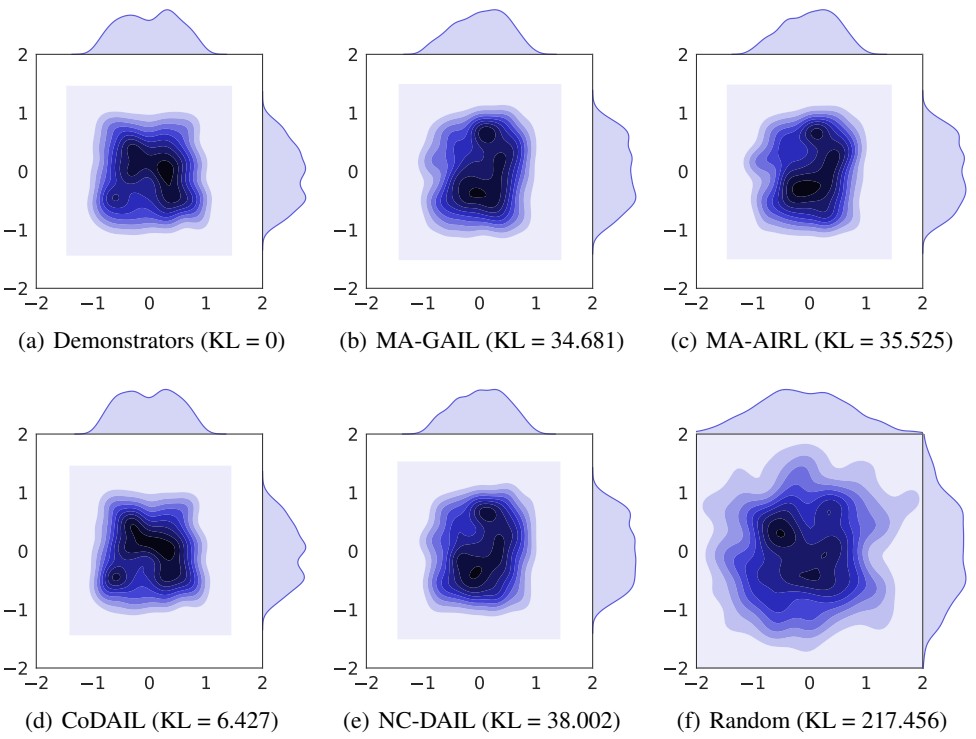

Figure 3: The density and marginal distribution of agents' positions, (x, y), in 100 repeated episodes with different initialized states, generated from different learned policies upon Cooperative-communication. Experiments are done under the same random seed, and we only consider one movable agent. KL is the KL divergence between generated interactions (top figure) with the demonstrators.

We show the density of interactions for different methods along with demonstrator policies conducted upon Cooperative-navigation in Fig. 4.

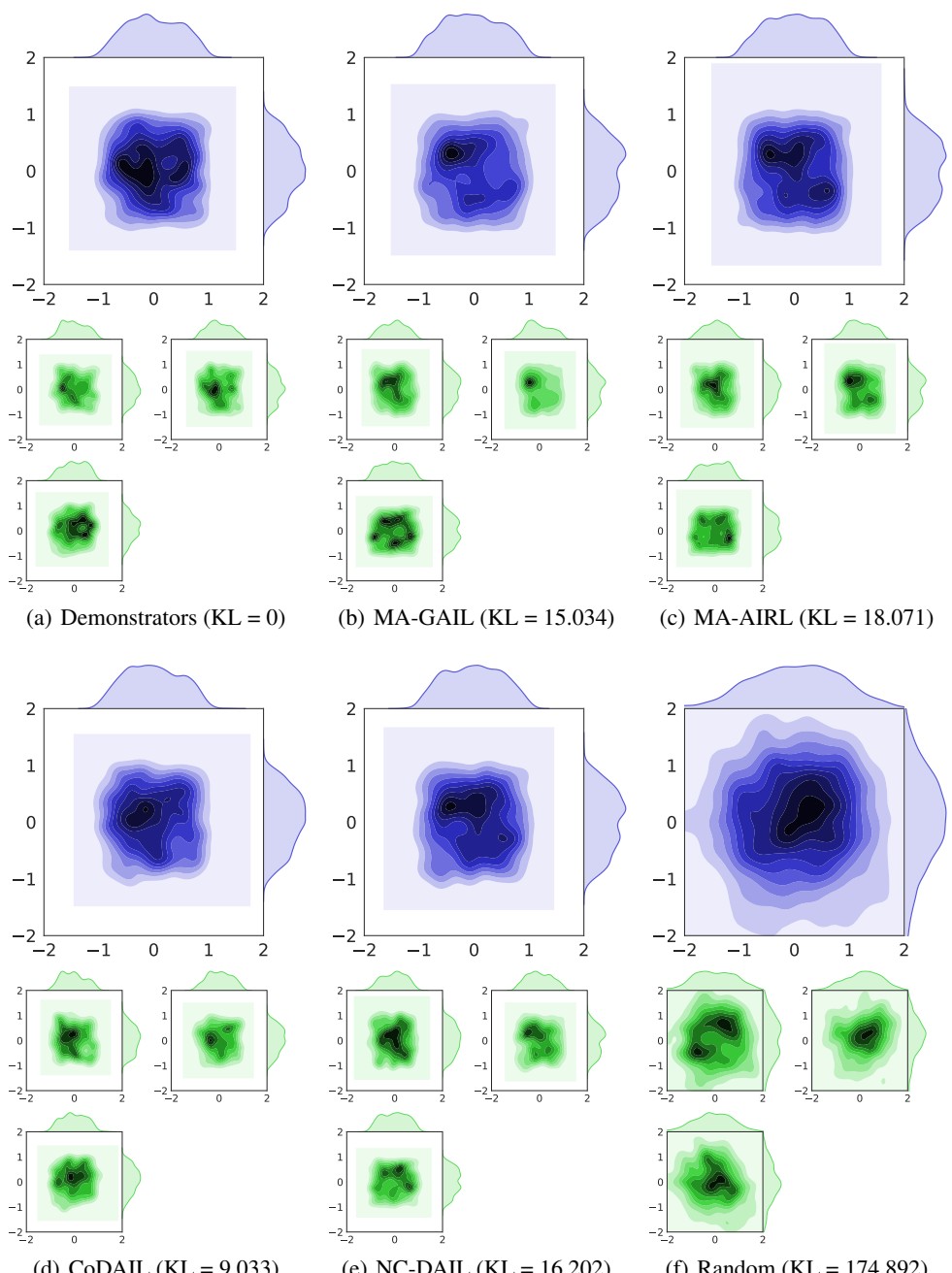

Figure 4: The density and marginal distribution of agents' positions, (x, y), in 100 repeated episodes with different initialized states, generated from different learned policies upon Cooperative-navigation. Experiments are done under the same random seed. The top of each sub-figure is drawn from state-action pairs of all agents while the below explain for each one. KL is the KL divergence between generated interactions (top figure) with the demonstrators.

We show the density of interactions for different methods along with demonstrator policies conducted upon Predator-prey in Fig. 5.

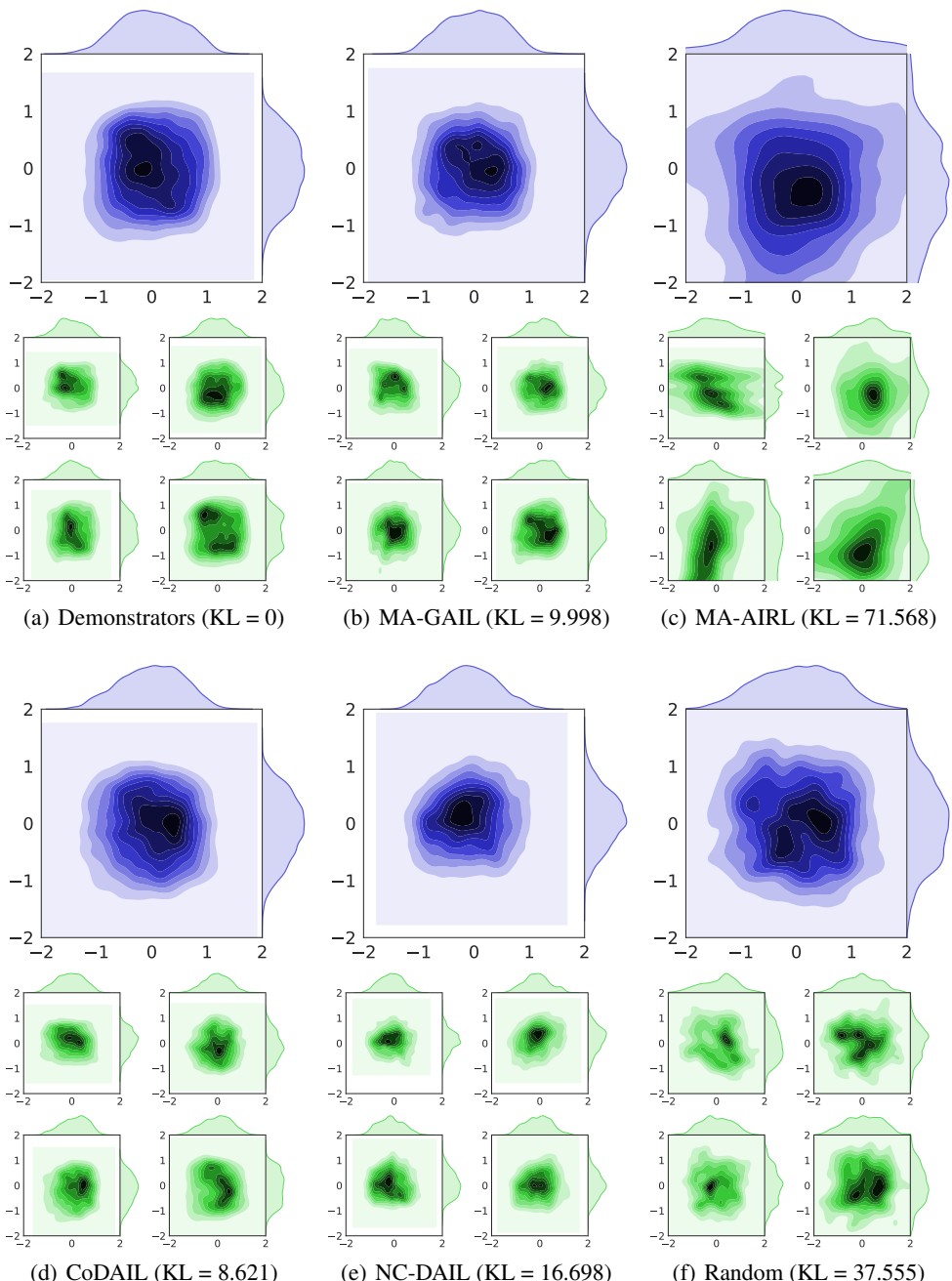

Figure 5: The density and marginal distributions of agents' positions, (x, y), in 100 repeated episodes with different initialized states, generated from different learned policies upon Predator-prey. Experiments are conducted under the same random seed. The top of each sub-figure is drawn from state-action pairs of all agents while the below explains for each one. The KL term means the KL divergence between generated interactions (top figure) with the demonstrators.

