# OpenReview forum: "Multi-Agent Interactions Modeling with Correlated Policies"
_ICLR.cc/2020/Conference — Accept (Poster)_

### Official Review · AnonReviewer2 · 2019-10-22
**Official Blind Review #2**

**Rating:** 8

**Review:**

In this work, a multi-agent imitation learning algorithm with opponent modeling is proposed, where each agent considers other agents’ expected actions in advance and uses them to generate their own actions. Assuming each agent can observe other agents’ actions, which is a reasonable assumption in MARL problems, a decentralized algorithm called CoDAIL is proposed. For each iteration of CoDAIL, (1) each agent trains opponent models (other agents’ policies) by minimizing either MSE loss (continuous actions) or CE loss (discrete actions), (2) samples actions from those opponent models, (3) updates individual rewards (discriminators) and critics and (4) updates policies with multi-agent extention of ACKTR (which is used in MA-GAIL and MA-AIRL as well).

The experiments in the submission show that there is a significant gain relative to baselines (MA-GAIL and MA-AIRL) in OpenAI Multiagent Particle Environments (MPE) in terms of (true) reward differences and KL divergence between agents’ and experts’ state distributions.

I think the empirical contribution of this work is clear to be accepted, but I give Weak Accept due to the following comments:

- I think there’s a similarity between Theorem 6 in MA-GAIL paper and Proposition 1 in the submission. I hope the difference between Proposition 1 and Theorem 6 to be clarified.

- Proposition 2 seems to me redundant because it’s neither important for theoretical analysis in 3.3 nor for the experiments. I believe a few sentences are enough to describe why authors choose \alpha=1 (or equivalent explanations).

- The authors suppose fully observable Markov Games in the paper, but it makes me confused when I consider the experiments in the submission. For example in Cooperative Navigation, each agent’s observation includes (1) position vector relative to agents and landmarks and (2) their own velocities (which cannot be observed by other agents directly). Since authors argue CoDAIL is a decentralized algorithm, I think agents are not allowed to use others’ observation for opponent modeling, but it seems that agents fully utilize others’ observations. I hope it to be clarified and if that’s the case, I wonder if we can regard CoDAIL as a decentralized method.

I’m willing to increase my score if my questions are clearly answered.

**Experience Assessment:**

I have published one or two papers in this area.

**Review Assessment: Checking Correctness Of Derivations And Theory:**

I carefully checked the derivations and theory.

**Review Assessment: Checking Correctness Of Experiments:**

I carefully checked the experiments.

**Review Assessment: Thoroughness In Paper Reading:**

I read the paper thoroughly.

---

> ### Author Response · Authors · 2019-11-07
> **Response**
>
> We sincerely thank you for your comprehensive comments on our paper and we carefully answer each of your questions as below.
>
> Q1. About Proposition 1
> Response:
> Theorem 6 of MA-GAIL paper and Proposition 1 of our paper appear to be similar though, they are technically different. MA-GAIL starts from a NE solution concept, while our deviation focuses on the objective of each agent, which makes the Proposition 1 in our paper more general and can be easily extended to different kinds of policy structures. Thus, only when every agent cares only the state without considering the others (independent, non-correlated policy structure), Proposition 1 of our paper collapses into Theorem 6 of MA-GAIL paper, where we feel free to add the objective of each agent in Proposition 1 as the total objective in Theorem 6.
> Besides, due to the "strict" NE setting, MA-GAIL must ignore the entropy item, which is not required in our $\epsilon$-NE setting. Thus, MA-GAIL can be regarded as a special case of our CoDAIL.
>
> Q2. About Proposition 2
> Response:
> Proposition 2 is a kind of simple and redundant. However, we feel it will make the paper more clear, because this is not a normally used importance sampling ratio of *occupancy measures* and one may be confused about the importance weight that whether it should be the ratio of two policies instead of two occupancy measures. Here we list it as an extra proposition to emphasize the technique of occupancy measure importance sampling.
>
> Q3. About the State/Observation Setting
> Response:
> Yes. This does exist and it also makes us confused when we read MA-GAIL and MA-AIRL papers, which care less about the partially observable property of Particle World environments. However, we show our understanding to interpret this problem as below.
>
> (1) First, either they or we do not consider to describe a PO setting because all of us wish to simplify the methodology and concentrate on the imitation learning architecture without caring about the partially observable settings.
>
> (2) Second, in most single-agent RL tasks, the normal inputs of agent policies are observations instead of states, e.g. the raw pixels of Atari games. However, in deep reinforcement learning (DRL), we can always map those observations into a low-dimensional latent state representation achieved by low-level layers of deep neural networks to achieve the function of state inference from observations in POMDPs, thus we usually care less about the observation/state in normal DRL tasks.
>
> (3) Thus we think that not only the previously mentioned works but many other MARL works who take Particle World as a MARL benchmark all stand at this point since the observations of Particle World contain comprehensive information to infer the latent states.
>
> (4) Since the other two works mainly conduct experiments on these Particle environments, at least we need to show the performance against baseline methods on Particle environments.

---

### Official Review · AnonReviewer3 · 2019-10-26
**Official Blind Review #3**

**Rating:** 6

**Review:**

The authors propose a decentralized adversarial imitation learning algorithm with correlated policies, which recovers each agent’s policy through approximating opponents action using opponent modeling. Extensive experimental results showed that the proposed framework, CoDAIL, better fits scenarios with correlated multi-agent policies.

Generally, the paper follows the idea of GAIL and MAGAIL. Differing from the previous works, the paper introduces \epsilon-Nash equilibrium as the solution to multi-agent imitation learning in Markov games. It shows that using the concept of \epsilon-Nash equilibrium as constraints is consistent and equivalent to adding the difference of the causal entropy of the expert policy and the causal entropy of a possible policy in RL procedure. It makes sense.

Below, I have a few concerns to the current status of the paper.

1.	The authors propose \epsilon-Nash equilibrium to model the convergent state in multi-agent scenarios, however, in section 3.1 the objective function of MA-RL (Equation 5) is still the discounted causal entropy of policy, the same as that of MA-GAIL paper. It is unclear how the \epsilon-NE is considered in modeling MA-RL problem.

2.	Rather than assuming conditional independence of actions from different agents, the authors considered that the joint policy as a correlated policy conditioned on state and all opponents’ actions. With the new assumption, the paper re-defines the occupancy measure and introduces an approach to approximate the unobservable opponents’ policies, in order to access opponents’ actions. However, in the section 3.2 when discussing the opponents modeling, the paper did not clearly explain how the joint opponent function \sigma^{(i)} is designed. The description \sigma^{(i)} is confusing.

3.	Typos: in equation 14 “i” or “-i”; appendix algorithm 1 line 3 “pi” or “\pi”.


**Experience Assessment:**

I have published in this field for several years.

**Review Assessment: Checking Correctness Of Derivations And Theory:**

I carefully checked the derivations and theory.

**Review Assessment: Checking Correctness Of Experiments:**

I carefully checked the experiments.

**Review Assessment: Thoroughness In Paper Reading:**

I read the paper thoroughly.

---

> ### Author Response · Authors · 2019-11-07
> **Response**
>
> We truly appreciate your helpful feedback.
>
> Q1. About "$\epsilon$-NE"
> Response:
> It is worth noting that we have shown our theoretical analysis in Sec 3.3 that the objective of our method in our derivation essentially equivalent to reaching an $\epsilon$-NE. However, this does not mean that we must begin with an $\epsilon$-NE solution: as shown in Eq. (6), we start by letting each agent achieve its own inverse RL objective to learn the expert policy $\pi$.
>
> To clarify the difference with MA-GAIL, our method theoretically shows a different perspective with MA-GAIL, which starts from a NE solution and a corresponding dual problem with ignoring the entropy term and does not solve the proposed dual problem. On the contrary, we show the multi-agent generative imitation learning problem (or multi-agent inverse reinforcement learning problem) can be seen to reach an $\epsilon$-NE solution concept, without limited in independent non-correlated policies and overlooking the entropy item. Thus, MA-GAIL can be regarded as a special case of our CoDAIL when ignoring the policy correlation among agents and the entropy item.
>
> Q2. About Opponents Modeling
> Response:
> We are sorry for the unclarity. In fact, the learning process of the joint opponent function $\sigma^{(i)}$  follows a normal way of opponent modeling.
>
> (1) Specifically, we construct a function $\sigma^{(i)}(a^{(-i)} | s): \mathcal{S} \times \mathcal{A}^{(1)} \times \cdots \times  \mathcal{A}^{(i-1)} \times  \mathcal{A}^{(i+1)} \times \cdots \times  \mathcal{A}^{(N)}\rightarrow {[0, 1]}^{N-1}$, as the approximation of opponents for each agent $i$.
>
> (2) Appendix B shows that in implementation: "Specifically for opponents models, we utilize a multi-head-structure network, where each head predicts each opponent's action separately, and we get the overall opponents joint action $a^{(-i)}$ by concatenating all actions.".
>
> (3) As Reviewer #2 says and shown in Eq. (17), opponent models are trained by minimizing either MSE loss (continuous actions) or CE loss (discrete actions).
>
> We have revised this part for clarity in the latest version of our paper.
>
> In sum, we think that at the current stage we have thoroughly answered your proposed questions, and according to your helpful suggestions, we have revised related parts for clarity in the latest version of our paper. Thus we sincerely wish you can re-consider and improve your rating for this work.

---

### Official Review · AnonReviewer1 · 2019-10-29
**Official Blind Review #1**

**Rating:** 6

**Review:**

This paper proposes to model interactions in a multi-agent system by considering correlated policies. In order to do so, the work modifies the GAIL framework to derive a learning objective. Similar to GAIL, the discriminator distinguishes between state, action, next state sequences but crucially the actions here are considered for all agents.

The paper is a natural extension of GAIL/MA-GAIL. I have two major points that need to be addressed.

1. The exposition and significance of some of the theoretical results is unclear.
- The non-correlated and correlated eqns in 2nd and 3rd line in eq. 8 are not equivalent in general, yet connected via an equality.
 In particular, Proposition 2 considers an importance weighting procedure to reweight state, action, next state triplets. It is unclear how this resolves the shortcomings of pi_E^{-1} being inaccessible. Prop 2 shifts from pi_E^{-1} to pi^{-1} and hence, the expectations in Prop 2 and Eq. 11 are not equivalent.
- More importantly, how are the importance weights estimated in Eq. 12? The numerator requires pi_E^{-1}, which is not accessible. If the numerator and denominator are estimated separately, it becomes a chicken-and-egg problem since the denominator is itself intended to be an imitating the expert policy appearing in the numerator?

2. Missing related work
There is a huge body of missing work in multi-agent interactions modeling and generative modeling. [1, 2] consider modeling of agent interactions via imitation learning and a principled evaluation framework of generalization in the Markov games setting. By sharing parameters, they are also able to model correlations across agent policies and have strong results on generalization to cooperation/competition with unseen agents with similar policies (which wouldn't have been possible if correlations were not modeled). Similarly, [3, 4] are other similar works which consider modeling of other agent interactions/diverse behaviors via imitation style approaches. Finally, the idea of correcting for the mismatch in state, action, next state triplets in Proposition 2 has been considered for model-based off-policy evaluation in [5]. They proposed a likelihood-free method to estimate importance weights, which seems might be necessary for this task as well (re: qs. on how are importance weights estimated?).

Re:experiments. Results look good and convincing for most parts. I don't see much value of the qualitative evaluation in Figure 1. If the KL divergence is low, we can expect the marginals to be better estimated. Trying out various levels of generalization as proposed in [2] would significantly strengthen the paper.

Typos
sec 2.1 Transition dynamics should have range in R+
Proof of Prop 2. \mu instead of u

References:
[1] Learning Policy Representations in Multiagent Systems. ICML 2018.
[2] Evaluating Generalization in Multiagent Systems using Agent-Interaction Graphs. AAMAS 2018.
[3] Machine Theory of Mind. ICML 2018.
[4] Robust imitation of diverse behaviors. NeurIPS 2017.
[5] Bias Correction of Learned Generative Models using Likelihood-free Importance Weighting. NeurIPS 2019.

**Experience Assessment:**

I have published one or two papers in this area.

**Review Assessment: Checking Correctness Of Derivations And Theory:**

I assessed the sensibility of the derivations and theory.

**Review Assessment: Checking Correctness Of Experiments:**

I assessed the sensibility of the experiments.

**Review Assessment: Thoroughness In Paper Reading:**

I read the paper at least twice and used my best judgement in assessing the paper.

---

> ### Author Response · Authors · 2019-11-07
> **** Response to Mathematics Details (1/3)**
>
> We sincerely thank the reviewer for the constructive comments.
>
> Q1: About "non-correlated and correlated eqs in 2nd and 3rd line in eq.8 are not equivalent yet connected via equality."
> Response:
> We are sorry for the confusion. By that we mean the joint policy can be decomposed into two different assumptions (either 2nd or 3rd line), we have revised the expression of Eq. (8) in our latest version.
>
> Q2: About "importance weight".
> Response:
> The main challenge to estimating the weight exactly is to estimate the (s, a) distributions of demonstrators' trajectories. Notice that the demonstrations are always insufficient for a low-variance estimation and it costs much to update such density estimations during training. In fact, we did have tried with KDE (kernel density estimation) to compute an "exact" importance weight but the results were not good. Thus we refer to [6] for a simple solution and in our paper, we have presented that "we fix $\alpha = 1$ in our implementation, and as the experimental results have shown, it has no significant influences on performance. Besides, a similar approach can be found in Kostrikov et al. (2018)." in the paragraph below Eq. (12).
>
> Reference:
> [6] Discriminator-Actor-Critic: Addressing Sample Inefficiency and Reward Bias in Adversarial Imitation Learning. I Kostrikov, KK Agrawal, D Dwibedi, S Levine. ICLR 2019.

---

> ### Author Response · Authors · 2019-11-07
> **** Response to "Missing related work." (2/3)**
>
>
> Q3: About "missing work in multi-agent interactions"
> Response:
> Thanks for the helpful suggestions. We have included some of those works of interactions modeling along with other opponent modeling papers to make it more clear in our latest version. In fact, we've read most of these works, yet we did not include them as they aim to address different problems.
>
> As we have formulated the problem of modeling multi-agent interactions from demonstrations as an imitation learning problem, we pay more attention to multi-agent imitation learning works as our comparable methods and the most related ones.
>
> Below we discuss each paper you mentioned in detail to clarify the differences between them and ours. Such discussions are also added to the related work of our latest version.
>
> 1 - [1] is the long paper of [2], which is an appealing work for modeling the among-agents interaction relationships as policy representations. Their problem setting has several important different points against us.
>
> 1.a - First, we focus on different tasks. They aim to learn the **representations function** of agent policies "based on their interactions", that is, to learn a "policies feature abstraction" with the latent relationships among agents rather than imitating their policies from demonstrations to regenerate similar interacted data with correlated policies. Their learned policy embedding function is able to characterize agent behaviors and can be used in kinds downstream tasks, which all take the policy embeddings as a core part, making it tough for us to try those generalization tasks since we only recover agents' policies.
>
> 1.b - Second, we consider different "comprehension" about interactions among agents. We care about the distribution of the overall interacted data sampled from correlated policies and how we can regenerating similar interacted data instead of analyzing the latent relationships among agents. Specifically, [1,2] regard interactions as the episodes that contain only k (in the paper they use 2 agents), which constructs an agent-interaction graph. That is, they focus on the latent relationships among agents.
>
> 1.c - Third, in [1,2], imitation learning is just a tool or technique to lean the policy embedding, which, by contrast, is the entire problem that we focus on.
>
> 1.d - Last but not least, parameter sharing is different from "correlated policy". Parameter sharing treats each agent as an independent individual to generalize the single-agent learning method in a multi-agent setting, which does not, in essence, consider the property of Markov Games and complicated "reasoning" policy. On the contrary, "correlated policy" means that each agent can infer about the others which explicitly considers opponents' policy in their decisionmaking process. See more details in [7,8,9]. In our setting, we want to model interactions considering such correlated policy structures, which is our motivation.
>
> 2 - The diverse behaviors of single-agent shown in [3] are different from the correlated interactions in a multi-agent setting. The main difference is that in single-agent setting one does not have to reason about the others, thus the generated trajectories are only related with the agent's own policy, which could be influenced by all agents in a multi-agent setting, and that's why the generated trajectories of all agents can be viewed as "interactions".
>
> 3 - [4] is a good work to model such a reasoning-like policy of agents, but they focus on MARL settings that interact with environments and learn policies with reward signals instead of an imitation learning setting that learning from pure demonstrations without reward signals (our task). Imitation learning is also just a technique to make use of the past trajectories of other agents. However, in future work, we can extend our work with their "theory of mind" policy structures to model complicated interactions.
>
> 4 - [5] cannot exactly solve the important weight problem. See details in our response to Q2.
>
> References:
> [7] Probabilistic recursive reasoning for multi-Agent reinforcement learning. Y Wen, Y Yang, R Luo, J Wang, W Pan. ICLR 2019.
> [8] A regularized opponent model with maximum entropy objective. Z Tian, Y Wen, Z Gong, F Punakkath, S Zou, J Wang. IJCAI 2019.
> [9] Opponent modeling in multi-agent systems. D Carmel, S Markovitch. IJCAI 1995.

---

> ### Author Response · Authors · 2019-11-07
> **** Response to Experiments (3/3)**
>
>
>
> Q4. About not "much value of the qualitative evaluation in Figure 1"
> Response:
> Figure 1 is a visualization of the distribution of trajectories of learned methods. As we can see in Figure 1(b) & 1(e), trajectories with similar KL-Divergence to the demonstrator trajectories do not necessarily have similar distribution patterns. This can be more clear in Figure 4(b) & 4(e). We show that our methods successfully generate *distribution-similar* trajectories against demonstrators more than just *KL-Divergence-better* methods.
>
> Q5. About "various levels of generalization"
> Response:
> We agree with your suggestion that it is better to consider more different-level evaluations. However, as shown in response 1.a of Q3, it is hard to straightly extend in [1,2]'s experimental settings for different tasks. And the major difficulty is that we learn the policy directly with no such a module as "policy embeddings" to achieve those downstream tasks.

---

### Author Response · Authors · 2019-11-07
**Overall Response - Motivations & Contributions**

We thank all reviewers for the valuable comments on improving the quality of this work and we would like to clarify our motivation and contributions:

1 - Motivation: In the real world, agents make decisions by constantly predicting and reasoning correlated intelligent agents' behaviors. We model such behaviors as correlated policy structure. Like in a driving scenario, a human driver makes decisions based on predicting and inducing the surrounding conditions that consisted of varies traffic participants; in a soccer game, a player would reason the next move of both his teammates and opponents before kick/moving decision.
In this paper, we aim to model the interactions among agents, by which we seek to perform high-fidelity simulation of the multi-agent environment with regenerating similar trajectories by imitating their correlated policies from demonstration data. However, traditional imitation methods such as GAIL, MA-GAIL and MA-AIRL lack the ability to model interactions from demonstrations sampled from these correlated policies.

2 - Contributions:
(1) We consider regenerating interacted trajectory data with recovered correlated policies, which is expected to follow a similar distribution with that from experts.
(2) We firstly propose to consider the influence of opponents in multi-agent imitation learning, in result showing the ability to learn from experts with correlated policies. With opponents modeling, our proposed framework CoDAIL gains the properties of decentralized-training and decentralized-execution.
(3) We show a different perspective that the multi-agent generative imitation learning problem (or multi-agent inverse reinforcement learning problem) can be seen to converge to an $\epsilon$-NE solution concept. Under our theoretical architecture, we start from the max-entropy inverse reinforcement learning objective of each agent while MA-GAIL paper derives from a NE solution and a corresponding dual problem. In result, MA-GAIL can be regarded as a special case of our CoDAIL when ignoring the policy correlation among agents.

According to your constructive comments, we have revised the equation symbols and discussions, fixed the typos and added more related works from different areas in our latest version paper, by which we think most confusions have been removed.

---

### Decision · Program_Chairs · 2019-12-19

**Decision:**

Accept (Poster)

**Comment:**

The paper proposes an extension to the popular Generative Adversarial Imitation Learning framework that considers multi-agent settings with "correlated policies", i.e., where agents' actions influence each other. The proposed approach learns opponent models to consider possible opponent actions during learning. Several questions were raised during the review phase, including clarifying questions about key components of the proposed approach and theoretical contributions, as well as concerns about related work. These were addressed by the authors and the reviewers are satisfied that the resulting paper provides a valuable contribution. I encourage the authors to continue to use the reviewers' feedback to improve the clarity of their manuscript in time for the camera ready submission.